# The Challenge of Developing a Test to Differentiate *Actinobacillus pleuropneumoniae* Serotypes 9 and 11

**DOI:** 10.3390/microorganisms13020280

**Published:** 2025-01-26

**Authors:** José Luis Arnal Bernal, Ana Belén Fernández Ros, Sonia Lacouture, Janine T. Bossé, László Fodor, Hubert Gantelet, Luis Solans Bernad, Yanwen Li, Paul R. Langford, Marcelo Gottschalk

**Affiliations:** 1Exopol. Polígono, río Gállego D14, 50840 San Mateo de Gállego, Spain; afernandez@exopol.com (A.B.F.R.); lsolans@exopol.com (L.S.B.); 2Research Group on Infectious Diseases in Production Animals and Swine and Poultry Infectious Diseases Research Centre, Faculty of Veterinary Medicine, University of Montreal, 3200 Sicotte, Saint-Hyacinthe, QC J2S 2M2, Canada; sonia.lacouture@umontreal.ca (S.L.); marcelo.gottschalk@umontreal.ca (M.G.); 3Section of Paediatric Infectious Disease, Imperial College London, St Mary’s Campus, London W2 1PG, UK; j.bosse@imperial.ac.uk (J.T.B.); yanwen.li@imperial.ac.uk (Y.L.); p.langford@imperial.ac.uk (P.R.L.); 4Microbiology and Infectious Diseases Department, University of Veterinary Medicine Budapest, Szent Istvan, No. 2, H-1078 Budapest, Hungary; fodor.laszlo@univet.hu; 5Research and Development Department, Ceva Biovac Campus, 49070 Beaucouzé, France; hubert.gantelet@ceva.com

**Keywords:** *Actinobacillus pleuropneumoniae*, serotype 9, serotype 11, *cpsF*, whole-genome sequencing (WGS), nanopore sequencing

## Abstract

*Actinobacillus pleuropneumoniae* is a major swine pathogen, classified into 19 serotypes based on capsular polysaccharide (CPS) loci. This study aimed to improve the diagnostic method to differentiate between serotypes 9 and 11, which are challenging to distinguish using conventional serological and molecular methods. A novel qPCR assay based on locked nucleic acid (LNA) probes was developed and validated using a collection of reference strains representing all known 19 serotypes. The assay demonstrated specificity in detecting the nucleotide variation characteristic of the serotype 9 reference strain. However, the analysis of a clinical isolate collection identified discrepancies between LNA-qPCR and serological results, prompting further investigation of the *cps* and O-Ag loci. Subsequent nanopore sequencing and whole-genome sequencing of a collection of 31 European clinical isolates, previously identified as serotype 9, 11, or undifferentiated 9/11, revealed significant genetic variations in the *cps* and O-Ag loci. Ten isolates had a *cpsF* sequence identical to that of the serotype 11 reference strain, while six isolates had single-nucleotide polymorphisms that were unlikely to cause significant coding changes. In contrast, 15 isolates had interruptions in the *cpsF* gene, distinct from that found in the serotype 9 reference strain, potentially leading to a serotype 9 CPS structure. In the O-Ag loci, differences between serotypes 9 and 11 were minimal, although some isolates had mutations potentially affecting O-Ag expression. Overall, these findings suggest that multiple genetic events can lead to the formation of a serotype 9 CPS structure, hindering the development of a single qPCR assay capable of detecting all *cpsF* gene mutations. Our results suggest that, currently, a comprehensive analysis of the *cpsF* gene is necessary to accurately determine whether the capsule of an isolate corresponds to serotype 9 or 11. Although such analyses are feasible with the advent of third-generation sequencing technologies, their accessibility, cost, and time to result limit their use in routine diagnostic applications. Under these circumstances, the designation of the hybrid serovar 9/11 remains a valid approach.

## 1. Introduction

*Actinobacillus pleuropneumoniae* is the causative agent of swine pleuropneumonia, a highly contagious disease that is a significant direct economic burden to the swine industry [1] and an indirect one through being a primary pathogen of the porcine respiratory disease complex [2]. Acute outbreaks of swine pleuropneumonia are characterized by fibrino-hemorrhagic and necrotizing pleuropneumonia, leading to high mortality rates. Furthermore, the chronic form of the disease can also negatively impact the industry by reducing daily weight gain, impairing feed conversion rates, and increasing medical costs [3,4].

Numerous methods have been developed to characterize *A. pleuropneumoniae* strains with the aim of studying their epidemiological distribution and, to some extent, predicting their virulence. Typically, characterization is by serotype, as it has a key role in determining immunological protection [3]. Serotype identification depends on antigenic and structural differences in capsular polysaccharides (CPSs) [5] or their corresponding CPS biosynthesis encoding genes. Currently, based on *cps* loci, a total of 19 serotypes have been described [6]. The antigenic determination of some *A. pleuropneumoniae* serotypes can be challenging since some serotypes share common or similar antigens. For example, serotypes 9 and 11, which are responsible for recurrent virulent outbreaks in several European countries such as Spain [7,8], Hungary [9], Italy [10], and Germany [11], are often indistinguishable and there is much interest in being able to differentiate the two serotypes. Firstly, differentiation would improve the epidemiological understanding of hybrid serotypes, especially given the limited information currently available. Secondly, and more importantly, differentiation would allow for improved disease control through an appropriate choice of bacterin vaccine, since protection is highly serotype-specific [12]. Since autogenous vaccines are now applied to entire production units, having precise epidemiological data to determine whether one or both serotypes are present could significantly enhance the efficacy of this control strategy.

Serotypes 9 and 11 were first described in the 1980s [13] and, several years later, their capsular antigens were defined [14], revealing only slight differences between the CPSs of these two serotypes. Serotype 9 capsular antigen repeating units consist of D-galactose and glycerol linked through phosphodiester groups, while serotype 11 additionally presents D-glucose as a residue of D-Galp [15,16].

Subsequent studies were conducted on the CPS biosynthesis genes of the serotype 9 (CVJ13261) and 11 (456153) reference strains. Complete *cps* loci sequences from both serotypes were aligned, identifying a single base insertion in the CVJ13261 *cpsF* gene as the unique distinguishing feature [17]. This insertion leads to an alternate start codon, resulting in a modification of open reading frame (ORF) length. The ORF for serotype 11 is 1242 base pairs (bp), in contrast to that of serotype 9, which is only 1146 bp. Thus, the proteins encoded by these different variants of the *cpsF* gene exhibit differences at the N-terminus, which may account for the subtle differences in CPS structure [15,16]. The functionality of the shorter version of the *cpsF* ORF remains unclear.

In addition to the separation of serotypes 9 and 11 by CPS biosynthesis genes, other approaches based on lipopolysaccharide (LPS) O-antigen (O-Ag) biosynthesis genes have been proposed, but these have also proven unsuccessful due to the similarity of their epitopes and respective genetic sequences [18,19]. Serological cross-reactions between serotypes, 1, 9, and 11 have been widely described [20]. When used to raise antibodies, the almost identical O-side chains of serotypes 1, 9, and 11 [18] can result in strong cross-reactive polyclonal antisera and monoclonal antibodies [21]. This limitation, along with the lack of standardized and commercially available reagents, has prompted diagnostic laboratories to adopt molecular techniques for serotype identification. Nevertheless, none of the molecular tests described to date can reliably differentiate between these serotypes, and the individual identification of serotypes 9 and 11 is a challenge for diagnostic laboratories. Notably, these two serotypes share an identical pore-forming RTX toxin profile [22,23], ApxI and ApxII [8,11], further complicating their individual identification. Thus, the true distribution of these serotypes remains uncertain, and most laboratories report isolates as “serotype 9/11”.

The advent of new technologies enables the evaluation of the genome in a substantial number of isolates previously identified as 9/11, with the aim of assessing the feasibility of accurate genotyping. One such technological advancement is the utilization of Locked Nucleic Acids (LNAs) for genotyping, which permits the discrimination of variation in individual nucleotides when employed in real-time quantitative PCR (Qpcr) [24,25]. An LNA-Qpcr-based approach allows for rapid results, cost efficiency, and the capacity to simultaneously analyze multiple samples consisting of microbiological isolates or animal samples such as lung tissues or oral fluids. However, it is important to note that the information obtained through LNA-based genotyping is limited to the specific target for which it is designed. In contrast, whole-genome sequence (WGS) determination provides a broader spectrum of information, encompassing the analysis of multiple complete genes. However, WGS is usually applied to microbiological isolates, is expensive, and does not meet the urgency typically required by clinical laboratories for timely diagnostic reporting. The aim of this study was to improve the diagnostic process of pleuropneumonia, offering a rapid and affordable method for veterinary diagnostic and research laboratories that identifies *A. pleuropneumoniae* as serotype 9 or 11.

## 2. Material and Methods

### 2.1. A. pleuropneumoniae Reference Strains and Field Isolates

A collection of reference strains representing the 19 serotypes described to date, along with two different collections of field isolates obtained from diseased animals in multiple countries, was assembled.

#### 2.1.1. *A. pleuropneumoniae* Reference Strains

This study included a collection of reference strains of the 19 known serotypes: serotype 1 (4074^T^), serotype 2 (S1536), serotype 3 (S1421), serotype 4 (M62), serotype 5 (L20), serotype 6 (Femø), serotype 7 (WF83), serotype 8 (405), serotype 9 (CVJ13261), serotype 10 (D13039), serotype 11 (456153), serotype 12 (8329), serotype 13 (N-273), serotype 14 (3906), serotype 15 (HS143), serotype 16 (A85/14), serotype 17 (16287) serotype 18 (73111555), and serotype 19 (A08-013).

#### 2.1.2. Field Isolates for LNA-Qpcr and *cpsF* Gene Sequencing

Additionally, 134 field *A. pleuropneumoniae* isolates, referred to hereafter as clinical collection A (“CCA”), obtained through routine microbiological procedures in their respective laboratories and identified as serotype 9, 11, or 9/11 by indirect hemagglutination (IHA) [9,26], PCR [6], or Qpcr [27], were included in this study (Table 1). Spanish isolates (*n* = 92) collected from respiratory outbreaks between 2018 and 2020 were identified using previously described commercial Qpcr kits [8,27], EXOone *A. pleuropneumoniae* (ref. APPL; Exopol, Spain) and EXOone APP serotype 9/11 (ref. AP09; Exopol, Spain), which target the *omlA* and *cps* genes, respectively. Isolates from France (*n* = 13), the Netherlands (*n* = 9), and Belgium (*n* = 8) were identified as serotype 9/11 using PCR [6]. Finally, isolates from Hungary (*n* = 12) were identified by IHA using polyclonal antibodies against *A. pleuropneumoniae* reference strains 1–18 generated in hyperimmunized rabbits [9,26]. Only those reactions were considered where the titer of hemagglutination reached that of the reference strain. Serotypes were determined based on the titers of homologous and cross-reactions with serotype-specific sera. Serotype 9 strains exhibited agglutination at a 1:1280 dilution with both serotype 9 and serotype 11 sera. In contrast, serotype 11 strains demonstrated strong agglutination at a 1:10,240 dilution with homologous serum, but only at a 1:640 dilution with hyperimmune serum raised against the serotype 9 strain. Additionally, serotype 11 strains showed weak agglutination with serotype 8, 12, and 15 sera. IHA identified seven Hungarian isolates as serotype 9 (327/12_A, 362/12_B, 448/12_C, 14/13_C, 79/13_B, 240/13_C, 391/13_B, and 155/14_B) and five as serotype 11 (161/15_D, 136/16_E, 150/16_F, 167/126_D, and 224/16_F).

#### 2.1.3. Isolates for Whole-Genome Sequencing (WGS)

Serotype 9 (CVJ13261) and 11 (456153) reference strains and a collection of 19 clinical isolates of *A. pleuropneumoniae,* collectively referred to as clinical collection B (“CCB”), previously identified serologically as serotype 9 (*n* = 3), serotype 11 (*n* = 4), or undifferentiated serotype 9/11 (*n* = 12), were selected for genomic analysis via WGS (see Table 2). These clinical isolates were generously provided by various collaborators and originated from six different countries between 1997 and 2019 (see Table 2). The inclusion criteria for these strains in the WGS study was that they were from diverse origins and operational, in that they were readily available to the research team dedicated to sequencing using the Illumina platform.

### 2.2. Molecular Studies

#### 2.2.1. Nucleic Acid Extraction

Genomic DNA from isolates was extracted using the commercial kit MagMAX™ CORE Nucleic Acid Purification Kit (Applied Biosystems, Foster City, CA, USA) following the manufacturer’s instructions through KingFisher Flex System (Thermo Fisher Scientific Inc., Worcester, MA, USA).

#### 2.2.2. Design of a Novel LNA-qPCR Test Differentiating Serotypes 9 and 11

Complete *cpsF* gene sequences from reference strains CVJ13261 (serotype 9) and 456153 (serotype 11), accession numbers ADOI00000000 and ADOK00000000, respectively, were obtained from NCBI GenBank. The *cpsF* sequences were aligned and compared using MSA software MAFFT (version 7) [28]. Two novel qPCR assays were then designed to differentiate serotype 9 from 11 within a multiplex format. Both reactions targeted the *cpsF* gene and shared the same forward and reverse oligonucleotide sequences. LNA technology was used to develop a specific probe for each serotype, 9 and 11, as indicated in Table 3. This discriminating multiplex qPCR was performed using 5 μL of template in a 20 μL reaction final volume using GoTaq™ Master Mix (Promega, Madison, WI, USA) in a 7500 FAST cycler (Applied Biosystems, Foster City, CA, USA). This LNA-qPCR ran under the following conditions: initial denaturation at 95 °C for 5 min, 40 cycles of denaturation at 95 °C for 15 s, and annealing and extension at 60 °C for 1 min. LNA-qPCR tests were validated using the reference strain collection and subsequently applied to analyze CCA isolates (*n* = 134).

#### 2.2.3. Sanger Sequencing of Partial *cpsF* Genes to Corroborate the LNA-qPCR Results

Alignment of the *cpsF* genes (Section 2.2.2) allowed for the design of a pair of primers to obtain partial *cpsF* sequences containing the single-nucleotide insertion [17]: SAN9/11seqF (5′-TGTTTAATGGCAAATAAGG-3′) and SAN9/11seqR (5′-TCGATAGTATTTGTTACACA-3′). The resulting 509 base pair amplicon comprised 173 nt and 336 nt upstream and downstream of the insertion point, respectively. Each PCR contained 10 μL (1–10 ng/μL) of genomic DNA extracted from each CCA isolate and 2.5 pmol of each primer in a 50 μL final reaction volume using the GoTaq™ Master Mix (Promega, Madison, WI, USA) in a 7500 FAST cycler (Applied Biosystems, Foster City, CA, USA). The PCR conditions were as follows: initial denaturation at 95 °C for 5 min; 40 cycles of denaturation at 95 °C for 15 s, annealing at 50 °C for 50 s, and extension at 72 °C for 45 s; followed by a final extension at 72 °C for 5 min. PCR products were purified, and both strands were Sanger-sequenced at a commercial facility (StabVida, Caparica, Portugal). The consensus sequences obtained were analyzed using the multiple sequence alignment software Unipro UGENE v50.0 [29].

#### 2.2.4. MinION Sequencing of Complete *cpsF* Genes

The complete sequences of *cpsF* genes from the 12 Hungarian isolates (CCA) were determined using the MinION platform from Oxford Nanopore Technologies (ONT, Oxford UK). For this purpose, 1.4 kb PCR amplicons generated using the ONT9/11seqF (5′-GACATACAGATATGGGAG-3′) and ONT9/11seqR (5′-GAACTCTTTATCGTAATTACTTAGC-3′) primers were sequenced using the Native Barcoding Kit 24 V14 (SQK-NBD114.24) following the manufacturer’s instructions. Nanopore sequencing was extended up to 24 h, and demultiplexing of the barcoded reads was performed using MinKNOW software (v.24.02.8) [30]. Reads with a Phred score > 9 were mapped to the *cps9F* sequence (ADOI00000000) from serotype 9 reference strain (CVJ13261).

#### 2.2.5. Whole-Genome Sequencing

Genomic DNA was extracted from the 19 *A. pleuropneumoniae* field isolates included in the CCB (Table 2) and from serotype 9 and 11 reference strains using a FastDNA SPIN kit (MP Biomedicals, Santa Ana, CA, USA). Draft WGS (MiSeq Illumina, San Diego, CA, USA), assembly, and Prokka annotations (v.1.14.5) [31] were performed by MicrobesNG (www.microbesng.com, access date: 23 January 2025) as described previously [32]. The resulting annotated draft assemblies were then examined using Artemis (Release 18.0.2) [33] to determine the location of the capsule (*cps*) locus, based on the identification of flanking *cpxD* and *ydeN* genes [17], and the LPS O-Ag biosynthesis locus by flanking *erpA* and *rpsU* genes [34].

## 3. Results

### 3.1. LNA-qPCR Results

The reference strain collection was used to validate the LNA-qPCR assays. All serotypes tested positive for the *A. pleuropneumoniae* qPCR (EXOone *A. pleuropneumoniae*; ref APPL). Both serotype 9 and 11 reference strains yielded positive results for the 9/11 qPCR assay (EXOone APP serotype 9/11; ref. AP09), while the novel LNA-qPCR assays tested positive exclusively for their respective serotype 9 and serotype 11 isolates; no positive results were obtained for any other strain from this collection. Therefore, the assays were considered analytically validated to proceed with the analysis of CCA field isolates (see Table 4).

All 134 CCA isolates tested positive for the serotype 11 LNA-qPCR (Cq values ranging from 10.2 to 29.01) and negative for the serotype 9 LNA-qPCR. Notably, all seven Hungarian isolates, previously identified serologically as serotype 9, tested positive in the serotype 11-specific LNA-qPCR assay and negative in the serotype 9-specific LNA-qPCR assay. The results are shown in Appendix A.

### 3.2. Sanger Sequencing of Partial cpsF Genes

*cpsF* partial sequences of all CCA isolates were identical with those from the serotype 11 (strain 456153) reference strain. Figure 1 shows the alignment of the partial *cpsF* sequence of a representative CCA field isolates (125691, 125842, 125941, 126195 and 137928) with those of the serotype 9 and 11 reference strains. All CCA isolates studied from Spain, France, the Netherlands, and Belgium had a single-nucleotide deletion at position 95 of the *cpsF* gene when compared with the serotype 9 reference strain (strain 456153).

### 3.3. MinION Sequencing of Complete cpsF Genes

The complete sequences of *cpsF* genes from the 12 Hungarian isolates from CCA were obtained using the MinION platform. Multiple-sequence alignment of the complete *cpsF* loci from the 12 Hungarian isolates and reference strains of serotype 9 and 11 is shown in Appendix A. Compared to reference serotype 9 sequence (CVJ13261), all 12 Hungarian isolates had the previously mentioned deletion at position 95, identical to the serotype 11 reference strain. In addition, several isolates had other mutations. Isolate 327/12_A had an insertion of a thymine (T) nucleotide at position 1096, resulting in an alternative ORF of 1200 bp. Isolates 362/12_B, 240/13_C, and 391/13_B had an 85 bp deletion from positions 408 to 493, leading to a truncated ORF of 513 bp. Furthermore, isolates 448/12_C, 14/13_C, and 79/13_B had an 11 nt (TAACTTAAAAT) insertion at position 536, resulting in a shorter ORF of 651 bp. Finally, isolates 161/15_D, 136/16_E, 150/16_F, 167/126_D, and 224/16_F had *cpsF* sequences identical to that of the serotype 11 reference strain (456153).

### 3.4. Whole-Genome Sequencing

Draft WGS were obtained for the 19 isolates of *A. pleuropneumoniae* serotype 9/11 from CCB described in Table 2 and their *cps* and O-Ag encoding genes identified as described in Section 2.2.5.

#### 3.4.1. *cps* Loci Analysis

Five isolates (1221B16, 10227, A05-01858-1_S10, A05-01858-3_S11, and Cz3_S6) had a *cpsF* sequence identical to that of the serotype 11 reference strain (456153). Additionally, six isolates (MIDG3457, MIDG3808, 20/4610, APP-TT911, 21/255, and 19/97-2) had SNPs relative to the *cps11F* reference strain sequence but would not be predicted to change the encoded protein. The remaining eight isolates had interruptions in their *cpsF* gene, but none were identical to the *cps9F* sequence found in the serotype 9 reference strain (CVJ13261). Internal stop codons led to various truncations, resulting in different sizes of ORFs. Czech A and 19192 isolates had 651 bp, MIDG3809 and 97/4080 isolates had 597 bp, 98/8106 and 99/4239 isolates had 756 bp, and MIDG3881 had 579 bp from the start of *cps11F*. The 6961_1_10 isolate had an interruption of the *cpsF* gene by an insertion sequence (IS) belonging to the IS*3* family [44], distinct from IS*Apl1* previously identified in *A. pleuropneumoniae* [45]. In this instance, 342 bp are upstream while 903 bp are located downstream of the insertion.

#### 3.4.2. O-Ag Loci Analysis

The sequence analysis of O-Ag loci in the serotype 9 and 11 reference strains did not reveal any relevant differences.

Examination of O-Ag loci from CCB field isolates (*n* = 19) identified that all but four have similar sequences to those of serotype 9 and 11 reference strains with occasional SNPs that are not predicted to affect the encoded proteins. However, four isolates have mutations that are predicted to impact O-Ag expression. Appendix A illustrates the distinct events associated with these mutations. Two isolates, MIDG3809 and APP-TT911, have mutations in the largest glycosyltransferase gene (locus tag D1102_05410, NCBI GenBank CP031865). MIDG3809 has a 5 bp deletion, while APP-TT911 has a 25 bp tandem duplication. Both mutations result in internal stop codons that are predicted to interfere with the expression of a functional protein. All isolates, except Cz3_S6, have the *rfbF* gene identical in sequence to that found in the serotype 9 (CVJ13261) and 11 (456153) reference strains with rhamnosyltransferase function. In Cz3_S6 isolate, a 122 bp deletion has resulted in the fusion of the *rfbF* gene with the downstream D1102_07830 gene (NCBI GenBank CP031865) encoding N-glycosyltransferase, resulting in a potentially non-functional single ORF of 1659 bp. Finally, 6961_1_10 contains multiple copies of insertion sequences with homology to IS*3* family that disrupt several loci. These include the glycosyltransferase gene (locus tag D1102_07820; NCBI GenBank CP031865) located upstream of the *rfbF* gene.

## 4. Discussion

The aim of this study was to provide diagnostic laboratories with a direct and simple test to differentiate between serotypes 9 and 11. The authors hypothesized that the single-nucleotide insertion observed in the *cpsF* gene of the serotype 9 reference strain (CVJ13261) [17] would be sufficient to distinguish between these two serotypes. For that purpose, a novel qPCR assay based on LNA probes was designed and proved effective in detecting the specific nucleotide variation described above. Validation of the assay involved a collection of reference strains from the 19 serotypes described to date. The proposed LNA-qPCR assay successfully detected only the targeted serotypes, confirming its analytical specificity.

Once the LNA-qPCR assays were validated, a collection of 134 *A. pleuropneumoniae* field isolates (CCA), previously identified as serotype 9/11, were analyzed using this new genotyping method. LNA-qPCR assays identified all CCA isolates, including those from Hungary that were previously identified as serotype 9 by serology (IHA), as serotype 11. Sanger sequencing of partial *cpsF* gene confirmed that all CCA isolates contained the specific deletion at position 95 of *cpsF*, corresponding to the serotype 11 reference strain. Complete agreement was observed between Sanger sequencing and the LNA-qPCR assay.

Thus, the molecular evidence available at that time indicated that all CCA isolates were serotype 11, with no detection of serotype 9 among those studied. However, inconsistencies were observed between the LNA-qPCR and Sanger results and the pre-existing serological data for the seven Hungarian CCA isolates, which had been originally identified as serotype 9 by IHA. The exact reasons for these discrepancies were not known at that moment. However, test-dependent cross-reactivity between serotypes 9 and 11 has previously been reported [46]. These cross-reactions were attributed to common epitopes present in the LPS O-chain of serotypes 1, 9, and 11, which were identified using monoclonal antibodies [21]. Given the lack of full concordance in the results and the absence of conclusive validation, the need for further analysis became evident. MinION sequencing of complete *cpsF* loci of Hungarian isolates from CCA and WGS of CCB isolates was conducted to identify new information that might permit the differentiation between the two serotypes.

The *cpsF* gene of serotype 9 and 11 reference strains comprise ORFs of 1146 bp and 1242 bp, respectively. The encoded protein is predicted to share the final 349 amino acids. However, the product of *cps9F* (glycosyltransferase) is likely non-functional since [5] the CPS of the serovar 11 reference strain contains β-D-Glpc as opposed to an acetyl group linked to de α-D-Galp in the backbone, as is found in the serovar 9 reference strain [14].

Significant changes in *cpsF* sequences of the Hungarian isolates (CCA) were identified by use of the MinION platform. The seven isolates previously identified as serotype 9 by serological methods had three distinct mutations. These included single-nucleotide insertion or substantial nucleotide deletions or insertions, resulting in ORF sizes of 1200 bp, 513 bp, and 651 bp, compared to the 1242 bp ORF of the serotype 11 reference strain. These alternative ORF sizes are unlikely to produce a functional enzyme required for incorporating the residue β-D-Glpc, which defines serotype 11 CPS, explaining why these isolates remained classified as serotype 9, consistent with the serological (IHA) data. In contrast, the five isolates serologically determined as serotype 11 had *cpsF* sequences identical to that of the serotype 11 reference strain.

The WGS analysis of CCB identified five isolates as having a *cpsF* gene sequence identical to that of the serotype 11 reference strain. Six isolates had SNPs that were not predicted to alter the encoded protein. Thus, all these 11 isolates would be predicted to produce a CPS identical to that of the serotype 11 reference strain. The eight remaining CCB isolates had various interruptions in their *cpsF* gene, none with the single-nucleotide insertion of the *cpsF* sequence of the serotype 9 reference strain [15]. Different ORF lengths were predicted, all likely to prevent the production of a functional glycosyltransferase. As a result, their CPS structure would likely resemble that of the serotype 9 reference strain. Notably, the insertion at position 95 in the *cpsF* gene of the serotype 9 reference strain appears to be a unique event, not observed in any of the other field isolates in this study.

The LPS structures of the serotype 9 and serotype 11 reference strains have subtle differences. The O-chain trisaccharide backbone of serotype 9 is only partially substituted by non-reducing 2-acetamido-2-deoxy-β-D-glucopyranosyl residues linked at the 1,3 position to the α-L-rhamnopyranosyl units in the main chain [5,47]. The analysis of O-Ag coding genes in the reference strains did not identify substantial differences that would lead to different predicted ORFs, aligning with the nearly identical LPS structures determined by Perry et al. [5]. Despite this, the genetic similarities found in this and other studies [48] may not hinder specific serotype identification through serological methods using polyclonal or monoclonal antibodies or the detection of infected animals using serological tests [49].

In addition, most of the CCB field isolates (*n* = 15) had well-conserved O-Ag loci. The mutations identified in the diverse O-Ag coding genes of four clinical isolates (MIDG3809, APP-TT911, Cz3_S6, and 6961_1_10) would be predicted to impact O-Ag synthesis and, potentially, serological results. This was not investigated further, but previous studies [38] suggest that strains with similar modifications are unlikely to induce a serological response detectable by LC-LPS ELISA [50]. Taken together, it is, therefore, not surprising that the serotype 1, 9, and 11 field isolates and reference strains show strong cross-reactivity using monoclonal or polyclonal antibodies raised against these serotypes. Considering the aforementioned factors, it should be emphasized that reference and field isolates do show strong cross-reactions both with both monoclonal and polyclonal antibodies during strain characterization, which corresponds to the inability to distinguish between serotypes 1, 9, and 11 in the serological diagnosis of infected animals [49].

Serologically, the sequence of the *cpsF* gene appears to be of major importance. For example, MIDG3457, MIDG3808, 1221B16, 10227, 20/4610, 21/255, A05-01858-1_S10, and A05-01858-3_S11 had a *cpsF* gene corresponding to serotype 11 but an O-Ag locus similar to that of serotype 9. Of these, MIDG3457, A05-01858-1_S10, and A05-01858-3_S11 were investigated serologically and were designated as serotype 11. Additionally, Cz3_S6 has a *cpsF* gene typical of serotype 11 but a significant disruption of two genes in the O-Ag system. Conversely, the isolates Czech A, MIDG3809, MIDG3881, 97/4080, 98/8106, 99/4239, 19192, and 6961_1_10 had truncated *cpsF* genes, likely to encode for a non-functional enzyme, leading to serotype 9 identification, exemplified by three of these isolates (97/4080, 98/8106, and 99/4239) being confirmed serologically by IHA as serotype 9.

Although the number of isolates for which the complete *cpsF* gene sequence was studied is limited—12 from CCA and 19 from CCB—they came from seven different European countries over two decades. Despite this relatively limited number, the genomic analysis revealed nine distinct mutation events at the *cpsF* locus that could lead to a serotype 9 designation when using IHA. Notably, as previously reported [51], the specific insertion found in the serotype 9 reference strain (CVJ13261) [17] was absent in all other field isolates analyzed, including those previously classified as serotype 9 by serological methods. While the novel LNA-qPCR method demonstrated specificity in detecting this insertion and was validated by Sanger sequencing, its use in clinical diagnostics is discouraged due to the rarity of this specific mutation [17] in clinical isolates. This limitation was not evidenced in previous molecular techniques which were intended to differentiate serotypes 9 and 11 [52] because diagnostic specificity validation was hindered by a lack of clinical isolate representation. In contrast, the present study incorporates a diverse range of field isolates, providing evidence that the serotype 9 CPS structure can be attributed to diverse *cpsF* truncations. Our collection only contained European isolates, and it remains to be determined whether the same or different mutations and/or insertions are present in isolates form other major pig production areas such as Australia, Brazil, Canada, and the USA.

As previously described [49], achieving reliable sero-diagnostics capable of differentiating between serotypes 9 and 11 through serum analysis remains highly challenging. Furthermore, the results obtained in this study indicate that multiple genetic events can give rise to the serotype 9 CPS structure instead of that of serotype 11. Consequently, unlike other assays developed for the identification of individual serotypes distinct from 9/11 and validated for direct use on clinical samples such as lung tissue or oral fluids [8,27], it is not feasible to design a single qPCR assay capable of detecting the diverse mutations observed in the *cpsF* gene for application directly to clinical specimens. Therefore, differentiating serotype 9 from serotype 11 based on the complete *cpsF* gene sequence requires *A. pleuropneumoniae* isolates obtained through bacteriological culture. This requirement significantly limits the applicability of this method, as microbiological isolation is typically restricted to clinical cases of pleuropneumonia, posing a substantial challenge for diagnostic laboratories when analyzing subclinical samples, such as oral fluids or tonsils.

The uncertainty associated with serological tests, combined with the absence of easily applicable molecular methods due to the lack of consistent genetic patterns, suggests that diagnostic laboratories with limited resources should continue designating these isolates as the previously proposed hybrid serotype 9/11. This classification remains a valid approach to encompass all isolates historically reported as serotype 9 or 11. Although the molecular techniques previously described for serotype 9/11 do not enable precise differentiation between these two serotypes, they remain highly valid, cost-effective, and relatively simple to implement. These procedures are valuable for both routine diagnostics and epidemiological studies, as they allow for the exclusion of both serotypes or, at a minimum, the confirmation of one. They can be applied to bacterial isolates [17] or directly to clinical samples [8,27]. Nonetheless, laboratories equipped with next-generation sequencing technologies can adopt the approach described in this study, i.e., obtain the sequence of the complete *cpsF* gene of isolates, as that allows for the precise identification of serotypes 9 and 11. Moreover, it is important to note that, although WGS provides comprehensive genetic information that could even reveal additional important genetic markers, its high cost and the need for specialized equipment may limit its accessibility for routine diagnostics in clinical settings, potentially hindering the widespread adoption of such advanced techniques in veterinary practice.

## 5. Conclusions

This study provides a novel characterization of multiple mutations in the *cpsF* gene of *A. pleuropneumoniae* field isolates previously identified as serotype 9/11, which are distinct from those found in serotype 9 and 11 reference strains. These mutations offer predictive insight into whether an isolate will produce a CPS typical of serotype 9 rather than serotype 11. Therefore, the use of advanced sequencing methodologies can enable differentiation between serotypes 9 and 11; however, its application is limited to the study of bacterial isolates and cannot be extended to other types of clinical or subclinical samples.

## Figures and Tables

**Figure 1 microorganisms-13-00280-f001:**
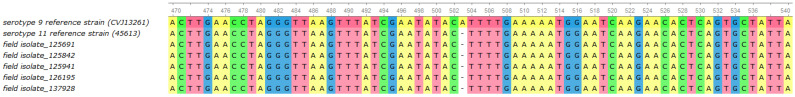
Multiple partial *cpsF* sequence alignment of serotype 9 (CVJ13261) and serotype 11 (45613) reference strains and representative field isolates from CCA: 125691, 125842, 125941, 126195, and 137928. These field isolates, along with the other CCA isolates (*n* = 134), did not exhibit the insertion [17] found in the serotype 9 (CV13261) reference strain. Unipro UGENE v50.0 software.

**Table 1 microorganisms-13-00280-t001:** Clinical collection A. Isolates of *A. pleuropneumoniae* serotype 9/11 studied by LNA-Qpcr and *cpsF* sequencing.

*N*	Origin	IHA ^1^	PCR ^2^	Qpcr ^3^	Molecular Techniques Used in This Study
92	Spain			9/11	LNA-Qpcr and partial *cpsF* sequencing (Sanger)
13	France		9/11		LNA-Qpcr and partial *cpsF* sequencing (Sanger)
9	The Netherlands		9/11		LNA-Qpcr and partial *cpsF* sequencing (Sanger)
8	Belgium		9/11		LNA-Qpcr and partial *cpsF* sequencing (Sanger)
7	Hungary	9			LNA-Qpcr and complete *cpsF* sequencing (MinION, ONT)
5	Hungary	11			LNA-Qpcr and complete *cpsF* sequencing (MinION, ONT)

This table shows the number, origin, and techniques previously used for serotype identification of the isolates included in CCA, as well as the techniques employed in the present study. ^1^ Indirect haemagglutination (IHA) test [9,26]; ^2^ conventional PCR described by Bossé et al. [17]; ^3^ commercial real-time PCR assay [8,27].

**Table 2 microorganisms-13-00280-t002:** Clinical collection B. Isolates of *A. pleuropneumoniae* serotype 9/11 studied by WGS.

Isolate	Year	Serotype	Origin	Collaborator	Institution
CVJ13261 (ref 9)	1985	9	The Netherlands	Marcelo Gottschalk	University of Montreal (Canada)
456153 (ref 11)	1987	11	The Netherlands	Marcelo Gottschalk	University of Montreal (Canada)
MIDG3457	2011	11	Cyprus	Andrew Rycroft	Royal Veterinary College (UK)
Cz3_S6	2008	11	Czech Republic	Katerina Nedbalcova	Veterinary Research Institute (Czech Republic)
Czech A	2008	9/11	Czech Republic	Katerina Nedbalcova	Veterinary Research Institute (Czech Republic)
A05-01858-1_S10	2005	11	France	Marcelo Gottschalk	University of Montreal (Canada)
A05-01858-3_S11	2005	11	France	Marcelo Gottschalk	University of Montreal (Canada)
10227	2011	9/11	France	Hubert Gantelet	Ceva Biovac Campus (France)
1221B16	2018	9/11	France	Hubert Gantelet	Ceva Biovac Campus (France)
97/4080	1997	9	Germany	Isabel Hennig-Pauka	University of Veterinary Medicine Hannover (Germany)
98/8106	1998	9	Germany	Isabel Hennig-Pauka	University of Veterinary Medicine Hannover (Germany)
99/4239	1999	9	Germany	Isabel Hennig-Pauka	University of Veterinary Medicine Hannover (Germany)
6961_1_10	2010	9/11	Germany	Judith Rohde	University of Veterinary Medicine Hannover (Germany)
19/97-2	2019	9/11	Germany	Isabel Hennig-Pauka	University of Veterinary Medicine Hannover (Germany)
21/255	2021	9/11	Germany	Isabel Hennig-Pauka	University of Veterinary Medicine Hannover (Germany)
MIDG3881	2016	9/11	Germany	Fabian Deutskens	Ceva Sante Animale (Germany)
APP-TT911	2019	9/11	The Netherlands	Tijs Tobias	University of Utrech (The Netherlands)
20/4610	2020	9/11	The Netherlands	Lydie van den Crommenacker	Dierenartsencombinatie Zuidoost (The Netherlands)
19192	2017	9/11	Poland		Bestvac
MIDG3809	2010	9/11	Poland	Fabian Deutskens	Ceva Sante Animale (Germany)
MIDG3808	2010	9/11	Poland	Fabian Deutskens	Ceva Sante Animale (Germany)

The year of isolation, previous serotype identification, geographical origin, and source of each strain are detailed.

**Table 3 microorganisms-13-00280-t003:** Oligonucleotide sequences for the LNA-qPCR multiplex assay for the detection of serotypes 9 and 11.

Code	Sequence	µM
APP9/11f	5′-CACGATGGATTTTCTCAAACTGAA-3′	500
APP9/11r	5′-TAGCCTTATCACCTAATAGCACTGA-3′	500
APP9p	/56-FAM/TTC + AA + AA + T+GT + A+TAT/3IABkFQ/	250
APP11p	/5HEX/TT + C+AA + AA + G+TA + TAT/3IABkFQ/	250

Locked nucleotides are indicated as “+N”.

**Table 4 microorganisms-13-00280-t004:** Serotype 9/11, serotype 9, and serotype 11 qPCR results of *A pleuropneumoniae* reference strains.

			^1^ *A*. *pleuropneumoniae*		^2^ 9/11		^3^ 9		^4^ 11	
Strain	Serotype	Reference	Result	Cq Value	Result	Cq Value	Result	Cq Value	Result	Cq Value
4074T	1	[26]	pos	16.62	neg		neg		neg	
S1536	2	[26]	pos	15.45	neg		neg		neg	
S1421	3	[26]	pos	15.04	neg		neg		neg	
M62	4	[26]	pos	16.51	neg		neg		neg	
L20	5	[26]	pos	14.93	neg		neg		neg	
Femø	6	[35]	pos	18.12	neg		neg		neg	
WF83	7	[35]	pos	14.3	neg		neg		neg	
405	8	[36]	pos	17.29	neg		neg		neg	
CVJ13261	9	[14]	pos	16.73	pos	16.42	pos	17.23	neg	
D13039	10	[37]	pos	17.27	neg		neg		neg	
456153	11	[13]	pos	15.33	pos	13.29	neg		pos	14.4
8329	12	[38]	pos	15.99	neg		neg		neg	
N-273	13	[39]	pos	16.86	neg		neg		neg	
3906	14	[40]	pos	17.05	neg		neg		neg	
HS143	15	[41]	pos	14.91	neg		neg		neg	
A85/14	16	[42]	pos	17.73	neg		neg		neg	
16287	17	[43]	pos	14.53	neg		neg		neg	
73111555	18	[43]	pos	14.75	neg		neg		neg	
A08-013	19	[6]	pos	15.02	neg		neg		neg	

“pos” means positive result due to Cq value < 38. “neg” means negative result due to no amplification or Cq value > 38. ^1,2^
*A. pleuropneumoniae* and hybrid serotype 9/11 commercial qPCR [27]; ^3,4^ Single serotype 9 and serotype 11 LNA-qPCR (current study).

## Data Availability

The original contributions presented in this study are included in the article/Appendix A. Further inquiries can be directed to the corresponding author.

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
