# Peer review of "The Challenge of Developing a Test to Differentiate Actinobacillus pleuropneumoniae Serotypes 9 and 11"

_microorganisms, 2025, doi:10.3390/microorganisms13020280_

Round 1

Reviewer 1 Report

Comments and Suggestions for Authors

Errors:

Line 176 error…

Line 250-253 alignment

Line 242 space

Line 339 number 1242 is underlined

Line 365 space

Several considerations and potential flaws must be noted:

1.       Validation of Assays: While the LNA-qPCR assays were validated using a reference strain collection, the article mentions discrepancies between qPCR results and pre-existing serological data. This raises questions about the reliability of the assays and suggests that further validation with a broader range of isolates may be necessary to confirm their accuracy.

2.       Cross-Reactivity Issues: The study acknowledges the potential for cross-reactivity between serotypes 9 and 11, which could lead to false positives or negatives in serotyping. This is a significant concern, as it may affect the interpretation of results and the subsequent management of infections.

3.       Cost and Accessibility of WGS: Although WGS provides comprehensive genetic information, its high cost and the need for specialized equipment may limit its accessibility for routine diagnostics in clinical settings. This could hinder the widespread adoption of such advanced techniques in veterinary practice.

4.       Sample Size and Diversity: The study analyzed a specific collection of 134 field isolates, which may not represent the full diversity of A. pleuropneumoniae strains in different geographical regions. A larger and more diverse sample size could provide more robust conclusions regarding the effectiveness of the diagnostic methods.

5.       It may be a bit strong to say that the article is focused on the "development of a test" in the traditional sense. Instead, it is more accurate to describe the research as an investigation into improving existing diagnostic methods for differentiating between Actinobacillus pleuropneumoniae serotypes 9 and 11. The study emphasizes the validation and application of new techniques, such as LNA-qPCR, rather than the creation of an entirely new test. Therefore, while the research contributes to the field of diagnostic testing, it is more about enhancing and refining methodologies rather than developing a completely new test from scratch.

Author Response

Reviewer 1

Line 176 error… A reference to Table 3 has been inserted (Line 191)

Line 250-253 alignment Title of Figure 1: Multiple partial cpsF sequence alignment is now placed before the figure (Line 264)

Line 242 space Removed as requested (Line 257).

Line 339 number 1242 is underlined Corrected (Line 361)

Line 365 space Removed as requested (Line 385)

Several considerations and potential flaws must be noted:

  1. Validation of Assays: While the LNA-qPCR assays were validated using a reference strain collection, the article mentions discrepancies between qPCR results and pre-existing serological data. This raises questions about the reliability of the assays and suggests that further validation with a broader range of isolates may be necessary to confirm their accuracy.

Discrepancies were observed between the results of the LNA-qPCR tests (instead of qPCR) and the pre-existing serological data when testing the Hungarian isolates previously identified as serotype 9 (n=7). Lines 340-342 have been slightly modified to clarify this point.

The 12 isolates from Hungary were the only ones previously identified using serological methods (IHA) due to the limited availability of laboratories equipped to perform such techniques.

It was validated that our LNA-qPCR assay accurately detected the mutation at position 95 of the cpsF gene for which it was designed. This was confirmed by comparison of LNA-qPCR results with those from Sanger sequencing of a considerably large collection of clinical isolates (CCA; n=134). However, the diagnostic validity of the LNA-qPCR assay was ruled out due to the rarity of the mutation event identified in the serotype 9 reference strain, which is not representative of the other serotype 9 field isolates under study.

  1. Cross-Reactivity Issues: The study acknowledges the potential for cross-reactivity between serotypes 9 and 11, which could lead to false positives or negatives in serotyping. This is a significant concern, as it may affect the interpretation of results and the subsequent management of infections.

Cross-reactivity issues are observed with previous qPCR/PCR tests or serological methods. However, the hybrid serotype 9/11 remains valid. A negative result for 9/11 confirms the absence of both serotypes 9 and 11. Conversely, a positive result cannot provide precise information regarding the specific serotype. This epidemiological challenge is discussed in lines 440-447. The novel contribution of this study lies in predicting the specific serotype (9 or 11) of isolates by sequencing the entire cpsF locus (lines 461-464).

  1. Cost and Accessibility of WGS: Although WGS provides comprehensive genetic information, its high cost and the need for specialized equipment may limit its accessibility for routine diagnostics in clinical settings. This could hinder the widespread adoption of such advanced techniques in veterinary practice.

We agree and the text has been modified to reflect this (line 453-457)  

  1. Sample Size and Diversity: The study analyzed a specific collection of 134 field isolates, which may not represent the full diversity of A. pleuropneumoniae strains in different geographical regions. A larger and more diverse sample size could provide more robust conclusions regarding the effectiveness of the diagnostic methods.

We acknowledge that our 9/11 results may not reflect those in major pig producing areas, e.g., Asia and the Americas. However, to our knowledge, we have assembled and done whole genome sequencing on the largest most diverse 9/11 collection ever described in the literature. Our results were enough to demonstrate that the LNA-qPCR technique is not valid in this context. It is highly unlikely that a larger European collection would yield a different conclusion for the LNA-qPCR.

A collection of strains was analyzed by sequencing the complete cpsF locus (using ONT or NGS), identifying nine distinct mutation events that could potentially classify an isolate as serotype 9 instead of 11. The geographical and temporal diversity among the field strains is significantly greater than that reported in previously published studies. (lines 409-411 and 418-421).

This is the first study to describe distinct mutation events at the cpsF level that differ from those observed in reference strains (line 409-411). If the collection of isolates subjected to WGS were larger, it is likely that additional mutation events leading to serotype 9 would be identified; nevertheless, the overall conclusion would remain unchanged: the study of cpsF mutations offers predictive insight into whether an isolate will produce a CPS typical of serotype 9 rather than serotype 11 (line 461).

However, in view of the comment, wehave added an extra sentence (lines 423-426 making to acknowledge that our results may not reflect those in other major pig producing areas. Indeed, we hope that our study will inspire researchers in those areas to determine the extent of variation in cpsF and O-Ag loci in their areas.

  1. It may be a bit strong to say that the article is focused on the "development of a test" in the traditional sense. Instead, it is more accurate to describe the research as an investigation into improving existing diagnostic methods for differentiating between Actinobacillus pleuropneumoniae serotypes 9 and 11. The study emphasizes the validation and application of new techniques, such as LNA-qPCR, rather than the creation of an entirely new test. Therefore, while the research contributes to the field of diagnostic testing, it is more about enhancing and refining methodologies rather than developing a completely new test from scratch.

Thank you for this comment. In fact, this study aims to enhance the diagnostic process for serotyping A. pleuropneumoniae by adapting a pre-existing technology based on LNA probes for qPCR. The text has been modified to address the comment (lines 27 and 121-122).

Reviewer 2 Report

Comments and Suggestions for Authors

This manuscript has discussed the challenges in developing a diagnostic test to differentiate between Actinobacillus pleuropneumoniae serotypes 9 and 11. These serotypes are difficult to distinguish using conventional methods. They have developed a novel qPCR assay using locked nucleic acid (LNA) probes and initially showed promise in differentiating between Actinobacillus pleuropneumoniae serotypes 9 and 11. However, analysis of clinical isolates revealed discrepancies between qPCR and serological results. Nanopore and whole-genome sequencing revealed significant genetic variations in the cps and O-Ag loci of clinical isolates previously identified as serotype 9 or 11. The manuscript doesn't explicitly state the severity of illness that is caused by serotype 9 or 11 and the clinical implications of misclassifying serotypes 9 and 11. This information is crucial to fully assess the significance of the problem and justify the research.

Specific comments:

1. The initial discrepancies between the LNA-qPCR results and serological typing raise concerns about the reliability of the initial identification of isolates. A clearer explanation and discussion of the discrepancies are needed.

2. I understand that the cpsF gene is important, while sole reliance on a single gene might overlook other genetic factors contributing to serotype variation, leading to incomplete understanding of the serotype 9/11 issue. A broader genomic analysis might reveal additional important genetic markers.

3. Please provide actual data of LNA-qPCR in detecting clinical samples.

4. Please correct Line 176. It should be table 3.2

Author Response

Reviewer 2

Comments and Suggestions for Authors

This manuscript has discussed the challenges in developing a diagnostic test to differentiate between Actinobacillus pleuropneumoniae serotypes 9 and 11. These serotypes are difficult to distinguish using conventional methods. They have developed a novel qPCR assay using locked nucleic acid (LNA) probes and initially showed promise in differentiating between Actinobacillus pleuropneumoniae serotypes 9 and 11. However, analysis of clinical isolates revealed discrepancies between qPCR and serological results. Nanopore and whole-genome sequencing revealed significant genetic variations in the cps and O-Ag loci of clinical isolates previously identified as serotype 9 or 11. The manuscript doesn't explicitly state the severity of illness that is caused by serotype 9 or 11 and the clinical implications of misclassifying serotypes 9 and 11. This information is crucial to fully assess the significance of the problem and justify the research.

We appreciate this point. Lines 71-78 have been added to develop more precisely the importance of differentiating between these two serotypes. Since there are no previous reports providing clear information about virulence differences in outbreaks caused by serotype 9 or 11, we chose to focus on developing control measures, particularly epidemiological studies and bacterin-based vaccines.

Specific comments:

  1. The initial discrepancies between the LNA-qPCR results and serological typing raise concerns about the reliability of the initial identification of isolates. A clearer explanation and discussion of the discrepancies are needed.

The text has been modified to specify where the inconsistencies were found: only in the results obtained by LNA-qPCR and IHA for the 7 Hungarian isolates serologically identified as serotype 9 (lines 340-343).

  1. I understand that the cpsFgene is important, while sole reliance on a single gene might overlook other genetic factors contributing to serotype variation, leading to incomplete understanding of the serotype 9/11 issue. A broader genomic analysis might reveal additional important genetic markers.

We agree and that would be an excellent topic for a future study but would require funding to be obtained. The focus of this study was on capsular polysaccharide (CPS) since it is widely accepted that this primarily determines serotype. Whether isolates are 9 or 11 has been difficult to determine, and of much interest in the community, and we have studied the most diverse 9/11 collection to enable a test to be developed. Biochemical differences in the molecular structure of serotype 9 and 11 CPS are observed in the D-Glucose residue (line 83). This residue is linked by a glycosyltransferase encoded specifically by cpsF (line 354-356). The different mutations described in this study may result in a non-functional glycosyltransferase, which would be unable to incorporate the D-Glucose residue. As a consequence, the antigenic structure of the CPS would resemble that of serotype 9 rather than 11.

Although this event would explain the differences between serotypes 9 and 11 and align with the available genetic and phenotypic information, the potential influence of genetic variability in other markers cannot be disregarded. We included a sentence to keep open this possibility (…”although WGS provides comprehensive genetic information that could even reveal additional important genetic markers”..). Lines 453-457.

  1. Please provide actual data of LNA-qPCR in detecting clinical samples.

The results of LNA-qPCR are now provided in supplementary file “Table 5”. See line 249.

  1. Please correct Line 176. It should be table 3.2

Corrected as requested. Line 191.

Reviewer 3 Report

Comments and Suggestions for Authors

The Challenge of Developing a Test to Differentiate Actinobacillus pleuropneumoniae Serotypes 9 and 11

In this study, the researchers aimed to create a test to distinguish between Actinobacillus pleuropneumoniae serotypes 9 and 11 by designing a locked nucleic acid (LNA) probe. They validated the probe using a collection of reference strains encompassing all 19 recognized serotypes. The findings indicate that a thorough analysis of the cpsF gene remains essential for accurately identifying whether an isolate's capsule belongs to serotype 9 or 11.

Comments

1.The study is well-structured, and the manuscript is clear and engaging as the authors utilized OpenAI's ChatGPT in the writing process.

2.The authors stated that they identified genetic variations in the cps and O-Ag loci across 9 and 11 serotypes. It is necessary to include illustrations of the sequencing alignments for these loci in the manuscript.

3. The authors have stated that serological methods were utilized in this study. Could the authors provide further details on how these methods were employed, including the specific techniques used, their application in the experimental workflow, and their contribution to the study's findings?

Author Response

Reviewer 3

In this study, the researchers aimed to create a test to distinguish between Actinobacillus pleuropneumoniae serotypes 9 and 11 by designing a locked nucleic acid (LNA) probe. They validated the probe using a collection of reference strains encompassing all 19 recognized serotypes. The findings indicate that a thorough analysis of the cpsF gene remains essential for accurately identifying whether an isolate's capsule belongs to serotype 9 or 11.

Comments

1.The study is well-structured, and the manuscript is clear and engaging as the authors utilized OpenAI's ChatGPT in the writing process.

Thank you very much for your consideration. AI assistance was used for syntax and language correction; however, the content is entirely original, and the authors are responsible for it.

2.The authors stated that they identified genetic variations in the cps and O-Ag loci across 9 and 11 serotypes. It is necessary to include illustrations of the sequencing alignments for these loci in the manuscript.

The alignment of ONT sequences from Hungarian isolates is particularly valuable, as these were the only isolates with IHA data available, enabling a direct comparison with genetic results.

We suggest that these genetic variations can be clearly demonstrated through a multiple sequence alignment figures 2a-2d of cpsF sequences. These figures are referenced in lines 272-274 of the manuscript and provided as a supplementary file (Figures 2a-2d). The file includes the 12 Hungarian isolates and reference strains of serotypes 9 and 11. The description of the genetic variations (lines 277-283) can now be easily followed with the aid of this figure. 

Additionally, Supplementary File Figure 3 has been included to illustrate the different mutation events identified throughout the loci belonging to the O-antigen (O-Ag) operon in the CCB isolates. The corresponding reference can be found in lines 308–309

  1. The authors have stated that serological methods were utilized in this study. Could the authors provide further details on how these methods were employed, including the specific techniques used, their application in the experimental workflow, and their contribution to the study's findings?

Serological methods have been limited in their use for studying the serotype of the Hungarian isolates (n=12) (lines 148-155). The IHA technique was employed as described in a previous publication by one of the co-authors (line 148).

The references provided are as follows:

Sárközi, R.; Makrai, L.; Fodor, L. Actinobacillus pleuropneumoniae serotypes in Hungary. Acta Vet Hung 2018, 66, 343–349

Kilian, M.; Nicolet, J.; Biberstein, E. L. Biochemical and Serological Characterization of Haemophilus pleuropneumoniae (Matthews and Pattison 1961) Shope 1964 and Proposal of a Neotype Strain; International Journal of Systematic and Evolutionary Microbiology 1978; Vol. 28

In order to clarify the issue as much as possible we have included some more information  about the criteria titration to determine the serotype by IHA: “Serotypes were identified on the basis of the titres of the homologous reactions and cross reactions with other serotype specific sera. Serotype 9 strains reacted in 1:1280 dilution both with type sera 9 and 11, while serotype 11 strains resulted agglutination in 1:10240 with the homologous serum but only 1:640 in the hyperimmune serum raised against type strain 9. Serotype 11 strains also gave weak agglutinations with serotype 8, 12 and 15.” (lines 148-155)

IHA has been added to the text so it is clear what serological method was used (Lines 335, 343, 364, 408).

The main contribution of the serological methods was providing information that did not agree with the results of the LNA-qPCR assay. This discrepancy led the authors to study the entire cpsF and O-Ag genes. (lines 340-343)

NB: Minor changes have also been made at the request of Dr Fodor. Specifically, they are:

Line 13. The affiliation of coauthor László Fodor has been slightly modified to be correct.

Line 480 The Hungarian funding source was inadvertently omitted and has been added.

Round 2

Reviewer 2 Report

Comments and Suggestions for Authors

No further commments.